# Inhalational Anesthetics Inhibit Neuroglioma Cell Proliferation and Migration via miR-138, -210 and -335

**DOI:** 10.3390/ijms22094355

**Published:** 2021-04-21

**Authors:** Masashi Ishikawa, Masae Iwasaki, Hailin Zhao, Junichi Saito, Cong Hu, Qizhe Sun, Atsuhiro Sakamoto, Daqing Ma

**Affiliations:** 1Department of Anesthesiology and Pain medicine, Graduate School of Medicine, Nippon Medical School, Tokyo 113-8603, Japan; masae-a@nms.ac.jp (M.I.); no1-saka@nms.ac.jp (A.S.); 2Division of Anaesthetics, Pain Medicine and Intensive Care, Department of Surgery and Cancer, Faculty of Medicine, Imperial College London, Chelsea & Westminster Hospital, London SW10 9NH, UK; hailin.zhao06@imperial.ac.uk (H.Z.); saitoj@hirosaki-u.ac.jp (J.S.); c.hu15@imperial.ac.uk (C.H.); q.sun17@imperial.ac.uk (Q.S.); 3Department of Anesthesiology, Graduate School of Medicine, Hirosaki University, Hirosaki, Aomori 036-8562, Japan

**Keywords:** microRNA, sevoflurane, desflurane, hypoxia inducible factor-1α, matrix metalloproteinase 9, glioma

## Abstract

Inhalational anesthetics was previously reported to suppress glioma cell malignancy but underlying mechanisms remain unclear. The present study aims to investigate the effects of sevoflurane and desflurane on glioma cell malignancy changes via microRNA (miRNA) modulation. The cultured H4 cells were exposed to 3.6% sevoflurane or 10.3% desflurane for 2 h. The miR-138, -210 and -335 expression were determined with qRT-PCR. Cell proliferation and migration were assessed with wound healing assay, Ki67 staining and cell count kit 8 (CCK8) assay with/without miR-138/-210/-335 inhibitor transfections. The miRNA downstream proteins, hypoxia inducible factor-1α (HIF-1α) and matrix metalloproteinase 9 (MMP9), were also determined with immunofluorescent staining. Sevoflurane and desflurane exposure to glioma cells inhibited their proliferation and migration. Sevoflurane exposure increased miR-210 expression whereas desflurane exposure upregulated both miR-138 and miR-335 expressions. The administration of inhibitor of miR-138, -210 or -335 inhibited the suppressing effects of sevoflurane or desflurane on cell proliferation and migration, in line with the HIF-1α and MMP9 expression changes. These data indicated that inhalational anesthetics, sevoflurane and desflurane, inhibited glioma cell malignancy via miRNAs upregulation and their downstream effectors, HIF-1α and MMP9, downregulation. The implication of the current study warrants further study.

## 1. Introduction

Surgery is the primary treatment of solid tumors world-wide. Sadly, the majority of patients still die due to cancer recurrence following surgery [1]. There are many risk factors related to cancer recurrence after surgery and one of those being considered is anesthetics/techniques [2]. Sevoflurane and desflurane are widely used inhalational anesthetics in clinical practice. Sevoflurane has been demonstrated to suppress cell proliferation, migration and invasion of glioma cells [3,4]. Recent studies have revealed that inhalational anesthetics can exert anti- or pro-cancer effects on cancer cell biology such as proliferation, migration, and apoptosis via some key factors including hypoxia inducible factor 1α (HIF-1α) [5] and Matrix metalloproteinases (MMPs) [4] depending on experimental conditions and cancer cell lines used. HIF-1α is an oxygen-labile DNA-binding transcriptional activator [6] and regulates cancer cell proliferation and invasion [7,8]. MMP9 plays a critical role in the process of tumor invasion [9] and progression [10]. Thus, it is important to explore the micro-mechanism of cancer biology changes after anesthesia exposure for establishing optimal anesthetic regimens for glioma surgery.

MicroRNAs (miRNA) are noncoding RNA molecules which negatively regulate protein expressions by post-transcriptional intervention to messenger RNAs (mRNA). miRNAs affect cancer cell metabolism, differentiation, proliferation, cell cycle control, apoptosis, invasion and migration in several cancer cell lines [11]. Anesthetics can alter a number of miRNA expressions in normal [12] and cancer cells [13]. However, the potential effects of anesthetics on cancer cell biology via miRNAs remain unknown.

The present study aims to investigate the effects of anesthetic sevoflurane and desflurane on glioma cell biology via miRNA expression changes. miR-138 [14] and -210 [15] regulate HIF genes, while miR-335 was reported to control MMP9 in glioma [16]. We hypothesized that inhalational anesthetics might modulate the HIF-1α and MMP9 cell signaling/pathway in glioma cells via miRNA expression changes.

## 2. Results

### 2.1. Sevoflurane and Desflurane Exposure Inhibited H4 Cell Proliferation and Migration

#### 2.1.1. Cell Migration

To examine the anesthetic effects on H4 cell biology, wound healing assay, Cell Counting Kit-8 (CCK8) assay and Ki67 staining analysis were performed up to 48 h after the exposure. In the wound healing assay, sevoflurane or desflurane exposure suppressed cell migration at 24 and 48 h after gas exposure (gap closure percentage at 24 h after anesthesia: sevoflurane 40.28 ± 1.35, *p* < 0.001; desflurane 41.46 ± 2.42, *p* < 0.001, vs. control 57.92 ± 1.05, the gap closure percentage at 48 h after anesthesia: sevoflurane 81.11 ± 1.43, *p* < 0.001; desflurane 81.22 ± 1.67, *p* < 0.001, vs. control 96.25 ± 1.11) (n = 6) (Figure 1a,b).

#### 2.1.2. Cell Proliferation Test

CCK8 assay analysis showed that sevoflurane and desflurane significantly decreased the cell proliferation at 24 h after gas exposure (cell proliferation relative to control: sevoflurane 0.92 ± 0.02, *p* < 0.001; desflurane 0.91 ± 0.02, *p* < 0.001, vs. control 1.00 ± 0.04) (n = 6) (Figure 1c). Both inhalational anesthetics reduced the Ki67-positive cells at 24 h after gas exposure (Ki67 positive cell percentage: sevoflurane 2.22 ± 0.23, *p* < 0.001; desflurane 2.16 ± 0.24, *p* < 0.001, vs. control 5.56 ± 0.41) (n = 6) (Figure 1d,e).

### 2.2. Sevoflurane Exposure Increased miR-210 Expressions and Desflurane Exposure Enhanced miR-138 and -335 Expressions

#### miRNA Changes after Anesthesia

miR-138 and -210 are known as regulators of HIF-1α and miR-335 is reported to control MMP9 expression. To determine their expression changes induced by sevoflurane or desflurane in H4 cells, qRT-PCR was performed just after gas exposure. The miR-138 and miR-335 expressions were upregulated by desflurane exposure (miR-138: sevoflurane 1.15 ± 0.23, *p* = 0.358; desflurane 1.46 ± 0.18, *p* = 0.001, vs. control 1.00 ± 0.11) (n = 6) (Figure 2f), (miR-335: sevoflurane 1.12 ± 0.27, *p* = 0.628; desflurane 1.90 ± 0.54, *p* = 0.001, vs. control 1.00 ± 0.11) (n = 6) (Figure 2h), whereas the expression of miR-210 (Figure 2g) was upregulated only by sevoflurane (sevoflurane 1.75 ± 0.37, *p* = 0.004; desflurane 1.02 ± 0.42, *p* = 0.992, vs. control 1.00 ± 0.14) (n = 6).

### 2.3. The Inhibition of miR-138, -210 and -335 Reversed the Sevoflurane- and Desflurane-Induced Suppression of Proliferation and Migration

#### 2.3.1. Cell Migration Ability after the Inhibitor Administration of miR-138, -210 and -335

To assess the effects of miRNA expression changes on cell activity, the miRNA inhibitors were transfected to H4 cells before anesthetic exposure. Without anesthetics, the inhibition of miR-138, -210 and -335 increased cell migration in the wound healing assay at 24 h after gas exposure (control + miR-138 inhibition 62.11 ± 0.85, *p* < 0.001; control + miR-210 inhibition 63.94 ± 3.75, *p* < 0.001; control + miR-335 inhibition 65.41 ± 3.57, *p* < 0.001, vs. control 54.03 ± 3.12) (n = 6) (Figure 2a,b).

#### 2.3.2. Cell Migration Ability after the Inhibitor Administration of miR-138, -210 and -335 with Sevoflurane Exposure

With sevoflurane exposure, only miR-210 inhibition increased cell migration at 24 and 48 h after anesthesia (24 h after the exposure: sevoflurane + miR-138 inhibition 43.12 ± 2.01, *p* = 0.768; sevoflurane + miR-210 inhibition 54.47 ± 3.57, *p* < 0.001; sevoflurane + miR-335 inhibition 45.75 ± 2.08, *p* = 0.057, vs. sevoflurane 41.32 ± 1.59) (n = 6) (Figure 2c,d), (48 h after the exposure: sevoflurane + miR-138 inhibition 87.44 ± 1.53, *p* = 0.870; sevoflurane + miR-210 inhibition 96.83 ± 3.83, *p* < 0.001; sevoflurane + miR-335 inhibition 90.53 ± 0.99, *p* = 0.061, vs. sevoflurane 85.86 ± 2.66) (n = 6) (Figure 2c,d).

#### 2.3.3. Cell Migration Ability after the Inhibitor Administration of miR-138, -210 and -335 with Desflurane Exposure

MiR-138 and miR-335 inhibition boosted gap closure at 24 and 48 h after desflurane exposure (24 h after the exposure: desflurane + miR-138 inhibition 50.01 ± 1.69, *p* < 0.001; desflurane + miR-210 inhibition 41.69 ± 2.14, *p* = 0.654; desflurane + miR-335 inhibition: 52.28 ± 2.84, *p* < 0.001, vs. desflurane: 40.08 ± 2.01) (n = 6) (Figure 2e,f), (48 h after the exposure: desflurane + miR-138 inhibition 91.24 ± 0.87, *p* < 0.001; desflurane + miR-210 inhibition 85.82 ± 2.40, *p* = 0.258; desflurane + miR-335 inhibition 96.75 ± 2.66, *p* < 0.001, vs. desflurane 84.22 ± 2.48) (n = 6) (Figure 2e,f).

#### 2.3.4. Cell Proliferation after the Inhibitor Administration of miR-138, -210 and -335

In Ki67 staining, the inhibition of miR-210 and -335 accelerated cell proliferation at 24 h after gas exposure (control + miR-138 inhibition 6.14 ± 0.19, *p* = 0.228, control + miR-210 inhibition 8.44 ± 0.41, *p* < 0.001, control + miR-335 inhibition 6.66 ± 0.41, *p* = 0.002, vs. control 5.59 ± 0.41) (n = 6) (Figure 3a,b).

#### 2.3.5. Cell Proliferation after the Inhibitor Administration of miR-138, -210 and -335 with Sevoflurane or Desflurane Exposure

With sevoflurane exposure, only miR-210 inhibition promoted Ki67 positive percentage (sevoflurane + miR-138 inhibition 1.89 ± 0.19, *p* = 0.054, sevoflurane + miR-210 inhibition 4.11 ± 0.21, *p* < 0.001, sevoflurane + miR-335 inhibition 2.24 ± 0.15, *p* = 0.999, vs. sevoflurane 2.23 ± 0.23) (n = 6) (Figure 3a,c), while miR-138 and miR-335 inhibition increased cell proliferation only by desflurane exposure (desflurane + miR-138 inhibition 4.49 ± 0.18, *p* < 0.001, desflurane + miR-210 inhibition 2.85 ± 0.14, *p* = 0.010, desflurane + miR-335 inhibition 4.33 ± 0.11, *p* < 0.001, vs. desflurane 2.16 ± 0.23) (n = 6) (Figure 3a,d).

### 2.4. Sevoflurane and Desflurane Exposure Attenuated HIF-1α Protein Expression Which Was Reverted by miR-138 and -210 Inhibitor Treatment

#### The Inhibition of miR-138 and -210 enhanced HIF-1α Expression at Any Conditions

HIF-1α is a key transcriptional factor in cancers which can be regulated by miR-138 and -210. Thus, HIF-1α expression was examined after miRNA inhibition and anesthetic exposure. In immunofluorescent staining, sevoflurane and desflurane exposure significantly decreased HIF-1α compared with the control (sevoflurane 0.78 ± 0.09, *p* < 0.008; desflurane 0.68 ± 0.15, *p* < 0.001, vs. control 1.00 ± 0.04) (n = 6). The inhibition of miR-138 and -210 enhanced HIF-1α expression (control + miR-138 inhibition 1.21 ± 0.14, *p* = 0.004, control + miR-210 inhibition 1.23 ± 0.08, *p* = 0.002, vs. control 1.00 ± 0.04; sevoflurane + miR-138 inhibition 1.33 ± 0.29, *p* < 0.001, sevoflurane + miR-210 inhibition 1.13 ± 0.19, *p* = 0.030, vs. sevoflurane 0.79 ± 0.09; desflurane + miR-138 inhibition 1.24 ± 0.14, *p* < 0.001, desflurane + miR-210 inhibition 1.44 ± 0.26, *p* < 0.001, vs. desflurane 0.68 ± 0.15) (n = 6) (Figure 4a,b).

### 2.5. Desflurane Exposure Attenuated MMP9 Protein Expression Which Was Reverted by miR-335 Inhibitor Treatment

#### The Inhibition of miR-335 Enhanced MMP9 Expression

MMP9 plays a main role in cancer cell migration, regulated by miR-335. MMP9 expression was examined after miRNA inhibition and anesthesia. In immunofluorescent staining, sevoflurane and desflurane exposure significantly decreased MMP9 compared with the control (sevoflurane 0.71 ± 0.16, *p* = 0.017; desflurane 0.69 ± 0.13, *p* = 0.012, vs. control 1.00 ± 0.18) (n = 6). The inhibition of miR-335 enhanced MMP9 expression with the presence of any anesthetics (control + miR-335 inhibition 1.66 ± 0.29, *p* < 0.001, vs. control 1.00 ± 0.18; sevoflurane + miR-335 inhibition 1.50 ± 0.09, *p* < 0.001, vs. sevoflurane 0.71 ± 0.16; desflurane + miR-335 inhibition 1.39 ± 0.10, *p* < 0.001, vs. desflurane 0.69 ± 0.13) (n = 6) (Figure 5a,b).

## 3. Discussion

This in vitro study with neuroglioma cells demonstrated that inhalational anesthetics exerted the anti-cancer effects via the miR-138/HIF-1α, miR-210/HIF-1α and miR-335/MMP9 pathways in this specific cancer cell line (Figure 6). miR-210 inhibitor attenuated the “anti-cancer” effect of sevoflurane and HIF-1α expression induced by sevoflurane, and miR-138 or miR-335 inhibitors reversed the “anti-cancer” effect of desflurane and HIF-1α or MMP9 expressions induced by desflurane. Sevoflurane decreased HIF-1α expression via miR-210, while desflurane downregulated HIF1-α and MMP9 expressions via miR-138 and miR-335, respectively. The current finding is not in line with our previous work in which showed that inhalational anesthetics had oncogenic effect via HIF-1a/miR-138 or -210 pathways in ovarian cancer (SKOV3) cells [17]. The “anti- or pro-cancer effects” of inhalational anesthetics via miR-138 or -210 are likely depending on cancer cell types.

Anesthetic/techniques may affect cancer cell biology such as proliferation, migration, invasion and apoptosis, and thus influence the clinical outcomes of cancer patients following surgery. Inhalational anesthetics have been reported to be capable of regulating gene expressions in human breast cancer and neuroblastoma [18], suggested to exert anti-proliferative effects in glioma cells [19] and decreased cell viability of non-small cell lung cancer cells [20]. Sevoflurane also has been reported to regulate multiple miRNAs in normal cells including brain [21] and peripheral organs [12,22] and in cancer cells, leading to cell biological changes. Sevoflurane can inhibit cell migration and invasion in colon cancer cells via the ERK/MMP9 pathway by regulating miR-203 [23]. Sevoflurane also attenuated glioma cell proliferation and invasion thorough the upregulation of miR-637 and miR-124, which suppressed the Akt1 expression and ROCK1 signal pathways, respectively [24,25]. miRNAs regulate cancer cell biology including cell proliferation and invasion in glioma [26] and ovarian cancer cells [27] via HIF-1α and MMP9 pathways. Several studies have demonstrated that HIF-1α can be regulated by miR-210 [15,28] and miR-138 [14] in melanoma cells. For example, miR-210 can act as a tumor suppressor, inhibiting cell proliferation in ovarian cancer [29] and laryngeal squamous cell carcinoma [30]. Conversely, miR-138 is downregulated in numerous cancers, including glioblastoma [31], and acts as a tumor suppressor gene by regulating the expression of various genes in many cancer types. For example, miR-138 suppressed cell proliferation, invasion and migration as a tumor suppressor gene in cervical cancer by targeting H2AX [32], and in ovarian cancer cells via SOX4 [33]. HIF-1α has been identified as one of the key regulators in tumor progression, cell proliferation, invasion and angiogenesis [34,35]. An increased expression of HIF-1α has been reported in a number of cancer cell lines [36], associated with tumor growth, metastasis, poor clinical prognosis [37], and chemoresistance [38]. HIF-1α expression may be increased via loss-of-function mutations of tumor-suppressor genes or gain-of-function mutations of oncogenes, and activation of the Phosphoinositide 3-kinase/Akt/Mammalian Target of Rapamycin (PI3K/Akt/mTOR) and Mitogen-activated Protein Kinase/Extracellular Signal-regulated Kinase (MAPK/ERK) pathways [39]. Inhalational anesthetics, e.g., Isoflurane, exposure increased HIF-1α expression in a concentration- and time-dependent manner, and HIF-1α translocated from the cytoplasm to the nucleus as a transcription factor, resulting in various downstream effectors being activated and then, in turn, increased proliferation and invasion in prostate cancer cells [5]. On the other hand, sevoflurane suppressed invasion and metastasis, and increases sensitivity to radiotherapy by inhibition of p38/MAPK, HIF-1α and MMP family proteins in non-small lung cancer cells [9,40]. Inhalational anesthetics accelerated cancer cell malignancy via HIF-1α/miR-138 or -210 in SKOV3 cells [17]. Together, all these studies may indicate that the effects of inhalational anesthetics on cancer cells via miR-138 or -210 were depending on cancer cell types.

MMPs are proteolytic enzymes that degrade components of the extracellular matrix (ECM) and basement membrane [41], contributing to cancer cell invasion and migration [42]. Increased MMP expression has been reported in many types of cancer cells [10,43], and its involvement in tumor progression [10], cell proliferation, remodeling, and invasion was reported in glioma cells [44]. MMP9 can also regulate vascular endothelial growth factor (VEGF) expression, an essential factor for angiogenesis [45] and tumor growth [46]. Furthermore, MMP9 expression in tumor tissue was correlated with the clinical tumor stages/grades and clinical outcome in glioma [47]. Sevoflurane and desflurane inhibited cell invasion through MMP9 downregulation in colorectal cancer cells [48]. miR-335 has been reported as a tumor invasion/metastasis suppressor in small cell lung cell lung cancer [49], gastric cancer [50] and ovarian cancer [51]. Downregulation of miR-335 in breast tumor tissue has been associated with overexpression of MMP2, MMP9 and VEGF and downregulation of tissue inhibitors of metalloproteinases (TIMP)1 and TIMP2 genes [52], which related to tumor size, cancer metastasis and cancer histological grades. This study showed the anti-cancer effect of desflurane but not sevoflurane via MMP9/miR-335.

In summary, our in vitro data showed that sevoflurane inhibited H4 cell migration and cell proliferation via miR-210/HIF-1α mechanism while desflurane via miR-138/ HIF-1α and miR-335/MMP9 modulation. However, this study is not free from limitations. Firstly, this is an in vitro study using one cell line. Each of the neuroglioma cell phenotypes have their own biological characteristics. Therefore, further investigations using other cell lines of neuroglioma are needed for further validation of the findings reported here. Secondary, the current experimental paradigm is only one relatively high concentration used with 2 h exposure time. Clinically, surgical time of brain tumor is sometimes longer than 2 h. Therefore, a variety of exposure time and concentration of these anesthetics are needed to determine their time- and concentration- effects. Furthermore, the experimental duration after gas exposure is relatively shorter (only up to 48 h) when considering postoperative recurrence. Lastly, the clinical value of the current study needs to be validated in vivo study and clinical settings.

## 4. Materials and Methods

### 4.1. Cell Culture

The human neuroglioma cell (H4) line (European Cell Culture Collection, Salisbury, UK) was cultured at 37 °C in a humidified atmosphere containing 5% CO_2_ balanced with air in Dulbecco’s Modified Eagle medium (Sigma-Aldrich, Dorset, UK), containing 10% fetal bovine serum (Thermo Scientific, Paisley, UK) and 1% penicillin (Sigma-Aldrich) for the experiments described below with or without inhalational agent exposures for further analyses.

### 4.2. Inhalational Anaesthetic Exposure

When cultures reached to 60%, they were exposed to the experimental gas mixture which consisted of 21% O_2_, 5% CO_2_ and either 3.6% sevoflurane or 10.3% desflurane balanced with N_2_ (BOC, South Humberside, UK) in a purpose-built 1.5 L airtight gas chamber, equipped with inlet and outlet valves. The chamber was placed in an incubator (Galaxy R CO_2_ chamber; New Brunswick Scientific, Enfield, CT, USA) at 37 °C for 2 h [53]. Other cohort cells were exposed to the same concentration gases without inhalational anesthetic served as controls. After exposure, cells were returned to the normal culture incubator until the further study.

### 4.3. Wound Healing Assay

Cells (2.5 × 10^4^) were seeded into each well (Culture-Insert 3 wells (Ibidi, Martinsried, Germany)), incubated for 24 h before wound was made and then exposed to the experimental gas mixtures as above for 2 h. The gap closure was monitored using an inverted microscope (CK30-SLP; Olympus, Tokyo, Japan) at 0, 24 and 48 h after gas exposure. Images were analyzed using Image J version 1.52a software (National Institute of Health, Bethesda, MD, USA). Gap closure (healing) was quantified according to the mean percentage remaining cell-free area compared with the area of the initial wound [54].

### 4.4. Cell Proliferation Test

Cells (7 × 10^3^) seeded into 96-well plates were exposed to the experimental gas mixture described above. The cell Counting Kit-8 (CCK8) reagent (Sigma-Aldrich) was added for 2 h before the readings of media were made at 450 nm using an ELx800 Microplate Reader (BioTek, Swindon, UK) at 24 h after gas exposure. Cell viability was expressed as a ratio relative to the controls.

### 4.5. RNA Extraction and Reverse Transcription

Immediately after the anesthetic exposure, the total RNA was extracted from cells using QIAzol Lysis Reagent and miRNeasy Mini Kit (Qiagen, West Sussex, UK) in accordance with the manufacturer instruction. The quantity and quality of RNA were assessed using a BioPhotometer (Eppendorf, Stevenage, UK). Samples with an A260/A280 ratio >1.8 were considered as to be sufficient quality for further analysis. RNA (1 ng) was converted to cDNA with miScript II RT Kit (Qiagen) using the thermal protocol of 37 °C for 60 min and 95 °C for 5 min in thermalcycler (Mastercycler^®^, Eppendolf, Stevenage, UK).

### 4.6. qRT-PCR

qRT-PCR was performed with miScript SYBR Green PCR Kit (Qiagen) and Rotor gene Q system (Qiagen). SNORD44 small nuclear RNA was used as an endogenous control. The commercial primers, miR-138-5p, miR-210-3p and miR-335-5p, were purchased from Qiagen. Thermocycle setting for RT-PCR was as follows; 95 °C for 15 min, 40 cycles of 94 °C for 15 s, 55 °C for 30 s and 70 °C for 30 s. Melting curve analysis was used to confirm the specificity of amplification. Expressions of miRNAs relative to SNORD44 were determined using the comparative 2^−ΔΔCt^ method.

### 4.7. miRNA Inhibitor Transfection

miR-138-5p (5′AGCUGGUGUUGUGAAUCAGGCCG), miR-210-3p (5′CUGUGCGUGUGACAGCGGCUGA) and miR-335-5p (5′UCAAGAGCAAUAACGAAAAAUGU) inhibitors (Qiagen) were purchased from Qiagen for transfection. miScript Inhibitor Negative Control (Qiagen) served as the negative control for miRNA inhibitor. Cells were transfected with 50 nM of each miRNA inhibitor or negative control using the HiPerFect transfection reagent (Qiagen) for 24 h. After transfection, the cells were exposed to anesthetics in the fresh medium.

### 4.8. Immunofluorescent Staining

Cells were fixed in 4% paraformaldehyde for 10 min and blocked with 10% normal donkey serum (Sigma-Aldrich) for 1 h, followed by overnight incubation at 4 °C with one of primary antibodies; rabbit polyclonal anti-Ki-67 antibody (1:500; Abcam plc, Cambridge, UK), rabbit polyclonal anti-HIF-1α antibody (1:300; Novus biologicals, Abingdon, UK), rabbit monoclonal anti-MMP9 antibody (1:300; Cell signaling Technology, London, UK). Cells were subsequently incubated in Alexa flour 568-conjugated secondary antibody (1:300; ThermoFisher scientific, Waltham, MA, USA) and co-stained with Vectashield mounting medium containing nuclear dye 4′, 6-diamidino-2-phenylindole–mounting medium (Millipore, Burlington, MA, USA). Cells were visualized with a BX60 wide-field fluorescence microscope (Olympus, Hamburg, Germany) under a 20× magnification. Images were captured using a cooled AxioCam camera (Zeiss, Oberkochen, Germany) with Zeiss software. Fluorescence intensity was quantified as the mean pixel intensity of relevant antibody staining using Image J version 1.52a software (National Institute of Health).

### 4.9. Statistical Analysis

All numerical data are presented as dot plots and expressed as mean ± SD. One-way analysis of variance followed by post-hoc Tukey’s test were used for data analyses using Prism version 8.0 (GraphPad Software, San Diego, CA, USA). A *p* value less than 0.05 was considered to be a statistical significance.

## 5. Conclusions

In conclusion, the present study showed that inhalational anesthetics inhibited cancer cell biology including cell proliferation and migration in neuroglioma cells via miRNA changes. Although both sevoflurane and desflurane provide “anti-cancer” effects through HIF-1α and MMP9 changes, the regulatory mechanisms of HIF-1α and MMP9 via miRNAs may be difference from one to the other inhalational anesthetics. This study may arguably contribute to the elucidation of the molecular mechanisms underpinning the effects of inhalational anesthetics on cancer cells. Our data may call more clinical studies to optimize anesthesia regimen for individual cancer type.

## Figures and Tables

**Figure 1 ijms-22-04355-f001:**
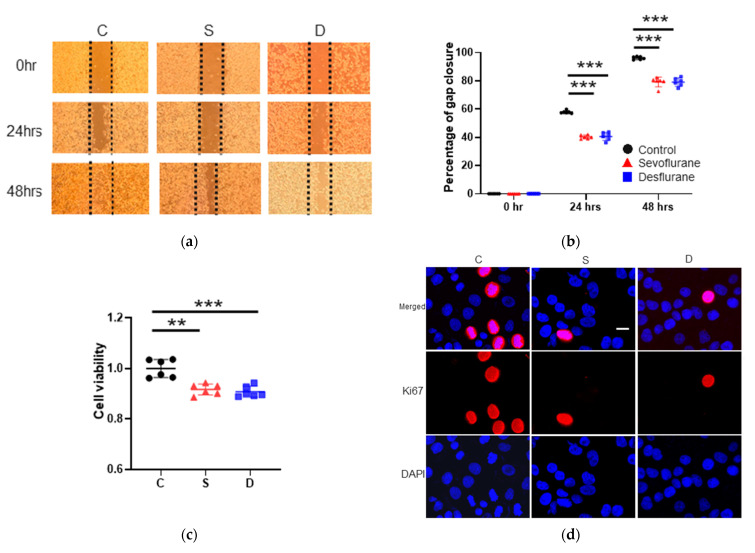
The changes of cell viability and miRNAs after inhalational anesthesia. (**a**) Neuroglioma (H4) cell migration analysis with wound healing assay after 2 h of inhalational anesthesia: control (left), 3.6% sevoflurane (middle) and 10.4% desflurane (right), at 0 h later (upper), 24 h later (middle) and 48 h later (bottom). The microscopic images at 0, 24 and 48 h after general anesthesia. (**b**) The comparison between anesthetics in percentage of gap closure by wound healing assay. (**c**) Cell proliferation analysis with CCK8 assay relative to control group. (**d**) Ki67 immunofluorescence staining: Ki67 (red), marker for cell proliferation, in control (left), sevoflurane- (middle) and desflurane-treated (right) H4 cells, counter-stained with DAPI (blue); x20 magnification, scale bar = 20 μm. (**e**) Comparison of percentage of Ki67 positive cells at 24 h after anesthesia exposure. (**f**–**h**) miRNA expressions evaluated with qRT-PCR compared to control group just after anesthesia exposure: (**f**) miR-138 (HIF-1α regulator), (**g**) miR-210 (HIF-1α regulator) and (**h**) miR-335 (MMP9 regulator). Data showed as plots and mean ± SD (n = 6). * *p* < 0.05, ** *p* < 0.01, *** *p* < 0.001; one-way ANOVA with Tukey-Kramer compared to the control group. C: control, S: sevoflurane, D: desflurane and CCK8: cell count kit 8.

**Figure 2 ijms-22-04355-f002:**
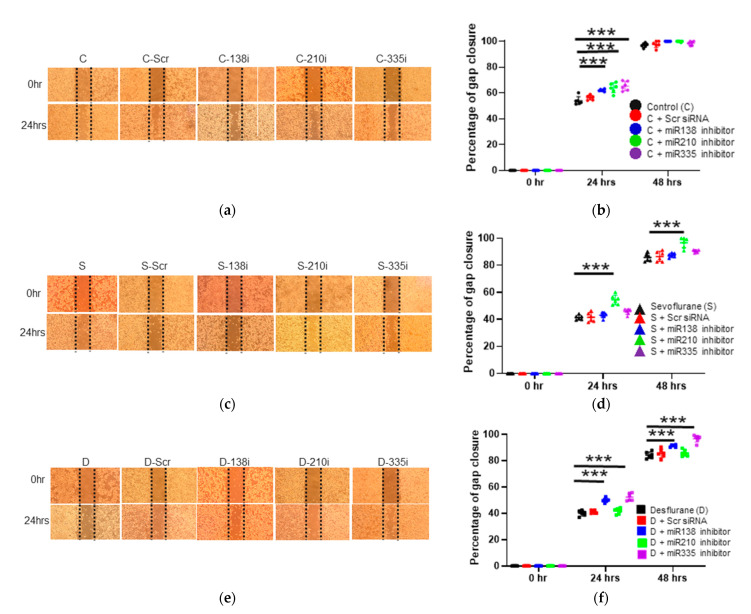
The changes of cell migration after inhalational anesthesia with miRNA inhibitor pretreatment. (**a**–**f**) Neuroglioma (H4) cell migration analysis with wound healing assay after 24 h and 48 h of inhalational anesthesia with miRNA inhibition pretreatment at 0 h (upper) and 24 h (bottom) after anesthesia. (**a**) The microscopic images after control anesthesia. (**b**) The comparison of gap closure percentage with control anesthesia and miRNA inhibitor pretreatment. (**c**) The microscopic images after 3.6% sevoflurane anesthesia with miRNA inhibitor pretreatment. (**d**) The comparison of gap closure percentage with 3.6% sevoflurane anesthesia and miRNA inhibitor pretreatment. (**e**) The microscopic images after 10.4% desflurane anesthesia with miRNA inhibitor pretreatment. (**f**) The comparison of gap closure percentage with 10.4% desflurane anesthesia and miRNA inhibitor pretreatment. Data showed as plots and mean ± SD (n = 6). *** *p* < 0.001. One-way ANOVA with Tukey-Kramer compared to each control group. C: control, S: sevoflurane, D: desflurane, Scr: scrambled miRNA, 138i: miR-138 inhibitor, 210i: miR-210 inhibitor and 335i: miR-335 inhibitor.

**Figure 3 ijms-22-04355-f003:**
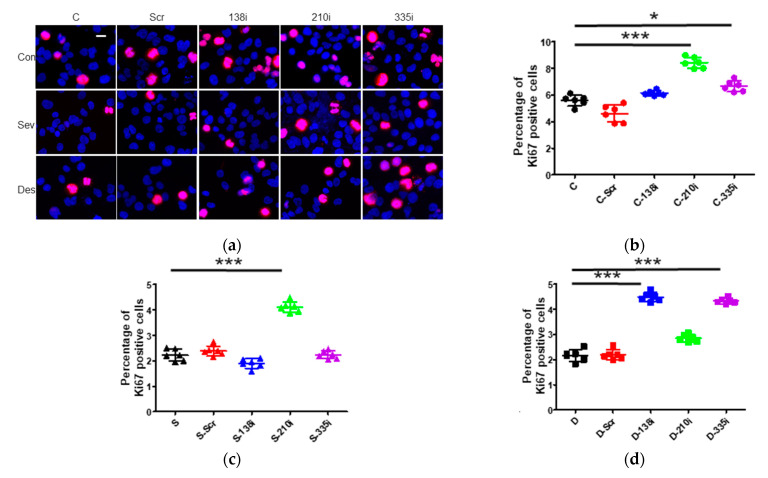
The changes of cell proliferation after 2 h of inhalational anesthesia with miRNA inhibitor pretreatment. (**a**) Neuroglioma (H4) cell proliferation analysis with Ki67 immunofluorescence staining: Ki67 (red), marker for cell proliferation, in control (upper), sevoflurane- (middle) and desflurane-treated (bottom) H4 cells, counter-stained with DAPI (blue); ×20 magnification, scale bar = 20 μm. (**b**–**d**) Cell proliferation analysis with Ki67 immunofluorescence staining with anesthesia and miRNA inhibitor pretreatment relative to each control group at 24 h after exposure: (**b**) control anesthesia, (**c**) 3.6% sevoflurane anesthesia and (**d**) 10.4% desflurane anesthesia. Data showed as plots and mean ± SD (n = 6). * *p* < 0.05, *** *p* < 0.001; One-way ANOVA with Tukey-Kramer compared to each control group. C: control, S: sevoflurane, D: desflurane, Scr: scrambled miRNA, 138i: miR-138 inhibitor, 210i: miR-210 inhibitor and 335i: miR-335 inhibitor.

**Figure 4 ijms-22-04355-f004:**
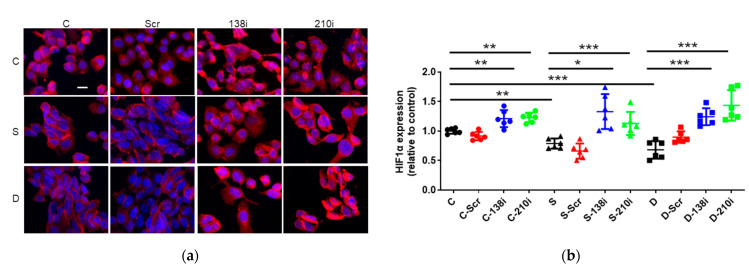
The HIF-1α expression changes after inhalational anesthesia with miRNA inhibitor pretreatment in neuroglioma (H4) cells. (**a**) HIF-1α immuno-fluorescence staining, HIF-1α (red) in control (upper), sevoflurane- (middle) and desflurane-treated (bottom) H4 cells with miRNA inhibitor pretreatment, counterstained with DAPI (blue); ×20 magnification, scale bar = 20 μm. (**b**) The comparison of the HIF-1α immunofluorescence intensity with each anesthetic and miRNA inhibitor pretreatment. Data showed as plots and mean ± SD. * *p* < 0.05, ** *p* < 0.01, *** *p* < 0.001, n = 6. One-way ANOVA with Tukey-Kramer compared to each control group. C: control, S: sevoflurane, D: desflurane, Scr: scrambled miRNA, 138i: miR-138 inhibitor and 210i: miR-210 inhibitor.

**Figure 5 ijms-22-04355-f005:**
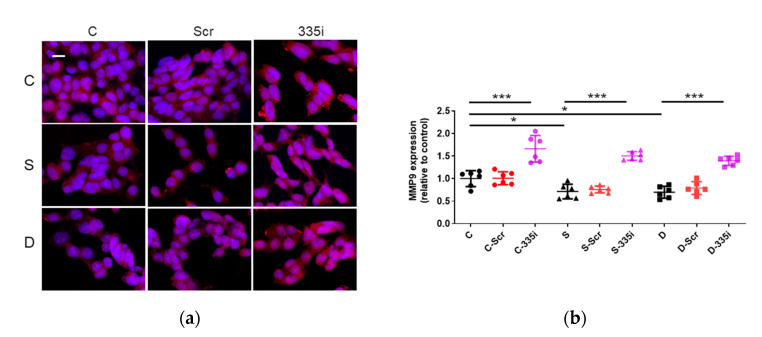
The MMP9 expression changes after inhalational anesthesia with miRNA inhibitor pretreatment in neuroglioma (H4) cells. (**a**) MMP9 immunofluorescence staining, MMP9 (red) in control (upper), sevoflurane- (middle) and desflurane-treated (bottom) H4 cells with miRNA inhibitor pretreatment, counterstained with DAPI (blue); ×20 magnification, scale bar = 20 μm. (**b**) The comparison of the MMP9 immunofluorescence intensity with each anesthetic and miRNA inhibitor pretreatment. Data showed as plots and mean ± SD (n = 6). * *p* < 0.05, *** *p* < 0.001; one-way ANOVA with Tukey-Kramer compared to each control group. C: control, S: sevoflurane, D: desflurane, Scr: scrambled miRNA and 335i: miR-335 inhibitor.

**Figure 6 ijms-22-04355-f006:**
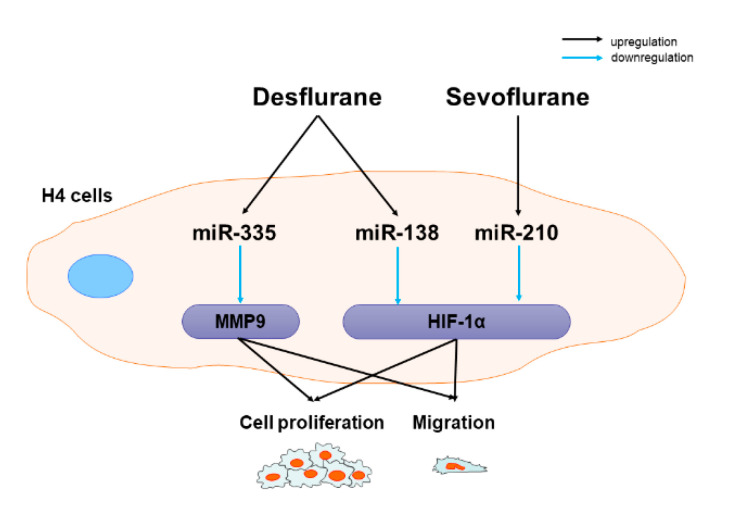
Inhalational anesthetics attenuated neuroglioma (H4) cell malignancy via upregulation of anticancer miRNA expressions. Sevoflurane exposure to H4 cells decrease H4 cell proliferation and migration via HIF-1α by miR-210 upregulation. Desflurane exposure to H4 cells decrease H4 cell proliferation and migration via HIF-1α and MMP9 by miR-138 and miR-335 upregulation, respectively.

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
