# Peer review of "Inhalational Anesthetics Inhibit Neuroglioma Cell Proliferation and Migration via miR-138, -210 and -335"

_ijms, 2021, doi:10.3390/ijms22094355_

Round 1

Reviewer 1 Report

Dear Authors, 

Congratulations on a very decent scientific piece of work. The article, although is only partly from my field, as I am an anaesthesiologist, I find very absorbing, and worth publishing. I wonder, if a control group with propofol, and extra group with isoflurane could be also possible - but that cannot be changed at that stage. 

When some bioethical concerns are attended to, that I described in section "Comments to Editors", I would recommend publishing it in a present form. 

Best Regards 

Author Response

Congratulations on a very decent scientific piece of work. The article, although is only partly from my field, as I am an anaesthesiologist, I find very absorbing, and worth publishing. I wonder, if a control group with propofol, and extra group with isoflurane could be also possible - but that cannot be changed at that stage.

When some bioethical concerns are attended to, that I described in section "Comments to Editors", I would recommend publishing it in a present form.

Response

Thank you very much for providing careful review. We are thankful for the time and energy you expended.

Reviewer 2 Report

The ms by Ishikawa et al. presents research data on the mechanism of glioma cell inhibition by inhalation anesthetics. However the whole idea is not novel, the presented research adds new details to the subject. Therefore in my opinion it is a valuable work to be published. However, it needs some clearances/corrections:

  1. Although the word "neuroglioma" is understandable it is used rarely compared to just "glioma". Obviously, that's how the producer of the H4 line describes the line, but that doesn't fit the current brain tumor classification and limits the clinical relations of the presented research. Therefore I suggest:
    1. In the text - using the name "neuroglioma" only when referencing to the cell line (line 36 "neuroglioma is a common type..." - that is simply not correct)
    2. Explain the choice of the cell line and the possible translation of the findings in that line to regular tumors' cell lines.
    3. Ideally, it would be to perform the same analysis on at least the second cell line (e.g. glioblastoma, which has a defined degree of malignancy).
  2.  The paper is about the influence of anesthetics on neuroglioma cells, with the possible clinical reference to glial tumors. Therefore the beginning of the Introduction should be more tailored to glial tumors. The lines 34-39 are too general, unnecessarily.
  3. Gas exposure of 2hrs is not bad, but a standard brain tumor surgery usually lasts longer (at least 3 hrs). Can authors explain the choice of the time period?
  4. The longest time of analysis after gas exposure was 48hrs. In the case of gliomas (and also other tumors) the inhibition of cancer cell proliferation and migration for 2 days is nothing. Why the authors didn't choose the longer time of observation (e.g. 7 days) to observe longer-lasting effects?
  5. Please perform one thorough review of the text - there are small punctuation, grammatical and spelling mistakes.

Author Response

Comment 1:

Although the word "neuroglioma" is understandable it is used rarely compared to just "glioma". Obviously, that's how the producer of the H4 line describes the line, but that doesn't fit the current brain tumor classification and limits the clinical relations of the presented research. Therefore, I suggest:

  1. In the text - using the name "neuroglioma" only when referencing to the cell line (line 36 "neuroglioma is a common type..." - that is simply not correct)
  2. Explain the choice of the cell line and the possible translation of the findings in that line to regular tumors' cell lines.
  3. Ideally, it would be to perform the same analysis on at least the second cell line (e.g. glioblastoma, which has a defined degree of malignancy).

Response 1:

We thank the reviewer for pointing out this. We agreed that only one subtype cell line of neuroglioma was used in the present study as one of the limitations, so we have clarified this in the discussion section (page 9, lines 270-272).

Comment 2:

The paper is about the influence of anesthetics on neuroglioma cells, with the possible clinical reference to glial tumors. Therefore, the beginning of the Introduction should be more tailored to glial tumors. The lines 34-39 are too general, unnecessarily.

Response 2:

Thank you for this comment. We have amended the introduction section accordingly (page 1, lines 34-36).

Comment 3:

Gas exposure of 2hrs is not bad, but a standard brain tumor surgery usually lasts longer (at least 3 hrs). Can authors explain the choice of the time period?

Comment 4:

The longest time of analysis after gas exposure was 48hrs. In the case of gliomas (and also other tumors) the inhibition of cancer cell proliferation and migration for 2 days is nothing. Why the authors didn't choose the longer time of observation (e.g. 7 days) to observe longer-lasting effects?

Response 3 and 4:

We thanked to the reviewer for pointing out. The experiment scheme including the exposure time and the observation period was based on our previous study (Oncotarget. 2016; 7:26042-26056.). We amended the reference (page 14, lines 497-498) and the discussion section (page 9, lines 272-278).

Comment 5:

Please perform one thorough review of the text - there are small punctuation, grammatical and spelling mistakes.

Response 5:

Thank you very much for pointing this. We have amended the manuscript accordingly.
